# Emotion is Not Just a Label:
# Latent Emotional Factors in LLM Processing

## Abstract

Large language models are routinely deployed on text that varies widely in emotional tone, yet their reasoning behavior is typically evaluated without accounting for emotion as a source of representational variation. Prior work has largely treated emotion as a prediction target, for example in sentiment analysis or emotion classification. In contrast, we study emotion as a latent factor that influences model processing and downstream question-answering performance. We analyze how emotional tone systematically alters attention geometry in transformer models, showing that metrics such as locality, center-of-mass distance, and entropy vary across emotions and correlate with downstream question-answering performance. To facilitate controlled study of these effects, we introduce Affect-Uniform ReAding QA (AURA-QA), a question-answering dataset with emotionally balanced, human-authored context passages. Finally, an emotional regularization framework is proposed that constrains emotion-conditioned representational drift during training. Experiments across multiple QA benchmarks demonstrate that this approach improves robustness on average across models and datasets, with particularly strong gains under distribution shift and improvements on several in-domain benchmarks.

## 1 Introduction

Research in natural language processing has long engaged with sentiment and emotion, spanning several areas. One prominent line investigates sentiment classification, in which algorithms categorize text by sentiment (Wankhade et al., 2022). Another examines how emotions are encoded within language models, blending mathematical and psychological perspectives (Reichman et al., 2025a; Zhang & Zhong, 2025). A third explores the emotional intelligence of large language models (LLMs) by presenting them with affective scenarios and examining their responses (Wang et al., 2023; Huang et al., 2024a; Zhao et al., 2023).

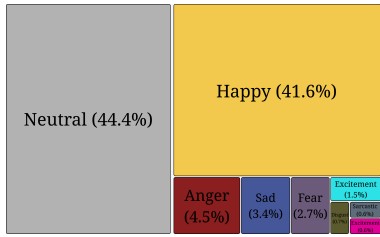

Figure 1: Distribution of emotions for a web corpus (Penedo et al., 2023).

Despite extensive work at the intersection of emotions and language processing, most prior work treats emotion as an object of prediction rather than a latent factor shaping processing. A text's tone shapes how readers interpret it. Because emotion and sentiment are integral to human experience, they are a natural part of writing. For instance, the line "They call me Professor and Doctor, forsooth, // For misleading many an innocent youth"[*] conveys very different meanings when read boastfully rather than in its intended doleful tone.

Most prior work asks language models to interpret tone directly, for example, to classify the mood of a stanza or to answer questions such as "What has made Faust upset?" While these studies treat emotion as an explicit signal, far less is known about how it influences ostensibly neutral reasoning tasks. Therefore, this study poses a different question: do variations in contextual tone influence performance on factual queries that are themselves non-emotional? For example, using context from earlier in the same passage, one might ask "What vocations did Faust study?" Does the passage's tonal misery affect the model's accuracy? If the passage were rendered cheerful instead, would performance improve?

---

[*]Faust, Part I

In the age of retrieval-augmented generation, LLMs increasingly process text drawn from diverse online sources, many of which exhibit varying emotional tone or subjective framing. To estimate the prevalence of emotional content on the internet[†], we analyze a 10-million-document subset drawn from the RefinedWeb dataset (Penedo et al., 2023), a large web-text corpus. Each document is labeled using the same emotion classifier described in Appendix A. Figure 1 shows that while most web text is neutral or happy, a substantial long tail of emotionally charged content remains that models must process robustly. Section 3 demonstrates that, on an emotionally balanced dataset, question-answering performance can differ by up to 12–13% between neutral and happy text. This paper investigates how such affective content influences model performance and proposes methods to improve robustness in these contexts.

To this end, this paper makes several contributions. First, it analyzes the model's attention geometry across different emotional contexts, arguing for the importance of studying the effect of emotions on LLM performance. Next it introduces an emotion-label-balanced dataset for question answering, AURA-QA, reducing the influence of emotional distribution imbalance when studying emotion-conditioned performance differences. Finally, a method for improving robustness to emotionally varied text is introduced. This method is evaluated across multiple QA benchmarks and improves robustness across models and datasets, with particularly strong gains under distribution shift.

## 2  Related Works

**Multi-Emotional Datasets.** Studying the effect of emotional context on QA performance requires datasets whose passages are annotated for emotion. However, to the best of our knowledge, the only natively constructed multi-emotional QA dataset currently available is entirely synthetic (Reichman et al., 2025c). Consequently, existing resources, TweetQA (tweets) (Xiong et al., 2019) and FriendsQA (dialogue) (Yang & Choi, 2019), were repurposed

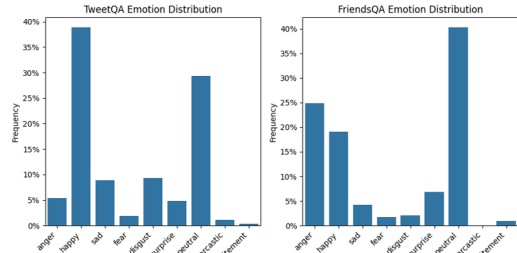

Figure 2: Emotion Distribution for TweetQA and FriendsQA.

and automatically labeled for emotional content using an automatic labeling pipeline (see Appendix A), as they naturally exhibit affective variation. As shown in Figure 2, these corpora display highly skewed emotional distributions, underscoring the need for a purpose-built dataset intentionally curated for balanced emotional diversity.

Related datasets such as SocialIQA (Sap et al., 2019), LongEmotions (Liu et al., 2025), EmotionBench (Huang et al., 2024b), and EmoBench (Sabour et al., 2024) probe social reasoning or emotional understanding, but do not test a model's ability to comprehend emotionally varied contexts when the question itself is emotion-neutral.

**Affective Reading Comprehension.** Prior work has largely assessed language models on emotional reasoning or emotional IQ—their capacity to infer or reason about affective states. Studies such as Liu et al., 2025 report only moderate competence in this domain, while structured prompting and reasoning chains yield partial gains in modeling social situations (Park et al., 2025). Other lines of research show that models can, with notable success, rediscover social interaction dynamics from first principles in goal-driven settings (Tejwani et al., 2023). Yet, when evaluated on Theory of Mind abilities, models still fall well short of human performance (Bortoletto et al., 2025).
In contrast, relatively little is known about how models interpret emotionally distinct contexts when the question itself is neutral. (Reichman et al., 2025c) introduces a synthetic dataset to explore this phenomenon, though its evaluation centers primarily on sarcasm. To date, there remains limited understanding of how emotional tone in retrieved passages modulates QA performance more broadly.

---

[†]For implementation details, see Appendix B.

# 3   Why Do Emotions Perform Differently?

| Feature | Definition | Interpretation |
|---|---|---|
| **Spatial Structure Features** | | |
| Center-of-Mass Distance (CMD) | $\dfrac{1}{H\lvert\mathcal{Q}\rvert}\sum_{h=1}^{H}\sum_{i\in\mathcal{Q}}\left\lvert i - \dfrac{\sum_j j A_{h,i,j}}{\sum_j A_{h,i,j}}\right\rvert$ | Measures how far attention extends from each query token; higher values indicate more non-local attention. |
| Tail Mass | $\dfrac{1}{\lvert\mathcal{Q}\rvert}\sum_{h=1}^{H}\sum_{i\in\mathcal{Q}}\sum_{j} A_{h,i,j}\,\mathbb{I}[\lvert i-j\rvert > d_0]$ | Quantifies reliance on long-range context by measuring attention mass beyond a fixed distance. |
| Locality | $\dfrac{1}{\sum_i m_i}\sum_{i,j} A_{h,i,j} m_i m_j \lvert i-j\rvert$ | Captures the expected spatial spread of attention; smaller values indicate tightly localized focus. |
| **Distributional Sharpness Features** | | |
| Key Entropy (KE) | $-\dfrac{1}{H}\sum_{h=1}^{H}\sum_{j} \bar{A}_{h,j} \log \bar{A}_{h,j}$ | Measures how uniformly attention mass is distributed across keys at the layer level. |
| Row Entropy (RE) | $-\dfrac{1}{H\lvert\mathcal{Q}\rvert}\sum_{h=1}^{H}\sum_{i\in\mathcal{Q}}\sum_{j} A_{h,i,j} \log A_{h,i,j}$ | Captures per-query diffuseness of attention; higher values indicate less decisive focus. |
| Top-1 Margin | $\dfrac{1}{H\lvert\mathcal{Q}\rvert}\sum_{h,i}\left(\max_j A_{h,i,j} - \max_{j\neq j^*} A_{h,i,j}\right)$ | Measures how decisively attention concentrates on a single token versus competitors. |
| Gini Coefficient | $\dfrac{1}{n}\left(n+1 - 2\sum_{j=1}^{n} j p_{(j)}\right)$ | Quantifies inequality in attention mass, emphasizing concentration rather than entropy. |
| **Depth-wise Dynamics** | | |
| Persistence | $\dfrac{1}{L-1}\sum_{\ell=1}^{L-1}\dfrac{\langle v^{(\ell)}, v^{(\ell+1)}\rangle}{\lVert v^{(\ell)}\rVert \lVert v^{(\ell+1)}\rVert}$ | Measures stability of attention patterns across layers; higher values indicate consistent focus through depth. |
| Curvature | $\dfrac{1}{L-2}\sum_{\ell=2}^{L-1}\lVert v^{(\ell+1)} - 2v^{(\ell)} + v^{(\ell-1)}\rVert_2$ | Captures how abruptly attention reallocates across layers, indicating volatility or "churn." |
| **Cross-Head Diversity** | | |
| Top-$k$ Overlap | $\dfrac{1}{B\binom{H}{2}k}\sum_{b,h<h'} \lvert S_{b,h} \cap S_{b,h'}\rvert$ | Measures redundancy in highly attended tokens across heads; lower values indicate specialization. |
| Head Similarity | $\dfrac{1}{\binom{H}{2}}\sum_{h<h'}\dfrac{\langle \hat{p}_h, \hat{p}_{h'}\rangle}{\lVert \hat{p}_h\rVert \lVert \hat{p}_{h'}\rVert}$ | Provides a continuous measure of correlation between heads' attention patterns. |
| **Task-Specific Focus** | | |
| Focus-To | $\dfrac{1}{\sum_i m_i}\sum_i m_i \sum_j A_{h,i,j} t_j$ | Measures how strongly valid queries attend to task-relevant regions (e.g., answer spans). |
| Focus-From | $\dfrac{1}{\sum_i t_i m_i}\sum_i t_i m_i A_{h,i,j}$ | Measures how attention originating from task-relevant regions is distributed outward. |

**Symbol key.** $A_{h,i,j}$ attention weight from query $i$ to key $j$ in head $h$; $H$ number of heads; $L$ number of layers; $\mathcal{Q}$ valid query positions; $m_i$ query mask; $t_i$ task indicator; $d_0$ distance threshold; $v^{(\ell)}$ layerwise attention summary; $S_{b,h}$ top-$k$ token set; $\hat{p}_h$ normalized attention vector.

Table 2: Summary of attention geometry features grouped by functional category.

When an off-the-shelf LLM answers questions based on contexts with different emotional valences, performance varies systematically across emotions (Table 1). This disparity persists both in zero-shot settings and when models are fine-tuned and evaluated on the target dataset. Results on a newly constructed dataset described in the next section, AURA-QA, is included. It demonstrates that these disparities remain even under controlled, emotionally balanced conditions, where differences in sample size cannot account for the effect. This observation motivates a mechanistic analysis of the performance gap from the perspective of attention geometry, specifically, how emotional tone alters the model's allocation of focus across tokens.

| Emotion | Results | |
|---|---|---|
| | Zero-shot | Trained |
| Neutral | 48% | 58% |
| Happy | 36% | 45% |
| Sad | 34% | 49% |
| Anger | 31% | 42% |
| Fear | 34% | 47% |
| Surprise | 36% | 54% |
| Disgust | 38% | 49% |
| Excitement | 39% | 55% |
| Sarcastic | 39% | 50% |

Table 1: Disparity in performance across emotions for LLaMA-3.1-8B on AURA-QA.

Attention is the primary mechanism by which a model allocates representational focus across tokens, governing how information is integrated across local and long-range contexts. To analyze how attention structure relates to question-answering performance and how emotional tone systematically alters this structure, we characterize attention using a set of geometric features summarized in Table 2. These features provide quantitative descriptions of how attention mass is distributed across the sequence, capturing locality, dispersion, and concentration patterns.

Although most of the attention features used in this work are not defined verbatim in prior literature, their design is motivated by recurring themes in earlier analyses of transformer attention. Spatial structure features are inspired by work showing that attentional distance and locality are central to transformer behavior, particularly in distinguishing local versus long-range information integration (Shaw et al., 2018; Beltagy et al., 2020). Relatedly, Tenney et al., 2019 analyze attention patterns using center-of-gravity–style measurements, motivating our center-of-mass distance formulation.

Distributional sharpness features are motivated by prior use of attention entropy to characterize concentration and diffuseness in attention distributions (Clark et al., 2019). Here, we extend this idea by separating entropy into row-wise and key-wise variants to distinguish per-query uncertainty from global key concentration. Depth-wise features are motivated by evidence that transformer layers exhibit systematic functional progression, with different depths specializing in different aspects of computation (Tenney et al., 2019; Rogers et al., 2020). This motivates explicitly measuring the stability and rate of change of attention patterns across layers. Cross-head diversity metrics are inspired by prior analyses of redundancy and specialization among attention heads, particularly in the context of head pruning (Michel et al., 2019). Finally, task-specific focus metrics are motivated by work probing how modifying or constraining attention distributions affects model behavior, suggesting that attention statistics conditioned on task-relevant regions can provide informative descriptive signals (Wiegreffe & Pinter, 2019).

## 3.1 Attention Geometry and Accuracy

To assess how attention geometry relates to task performance, we trained logistic regression models to predict question–answering accuracy using attention-derived features alone. For each input, attention statistics were computed at every transformer layer and attention head, summarized across heads within each layer using fixed summary operators (mean, standard deviation, and upper and lower quantiles), and then averaged across layers to obtain a single feature vector per example. Scalar per-layer metrics were treated analogously and included directly in the layerwise averaging. All features were standardized using z-scoring prior to model fitting. Model performance was evaluated using 5-fold stratified cross-validation with an L2-regularized logistic regression, and reported using ROC–AUC.

Using the full aggregated feature set jointly yielded an average AUC of 0.75 ± 0.03, indicating that attention geometry captures a substantial fraction of accuracy variance. To assess the predictive strength of individual metrics, we additionally trained univariate logistic regression models using each feature independently under the same cross-validation and standardization protocol. The strongest individual predictors were focus-from top-k mass features (AUC = 0.74 ± 0.01), computed from summary statistics of the distribution of attention emitted from answer-span tokens. Higher performance was associated with lower focus-from entropy and greater concentration of attention mass on the top-k targets, indicating that successful predictions tend to coincide with sharply constrained outward attention from semantically critical regions.

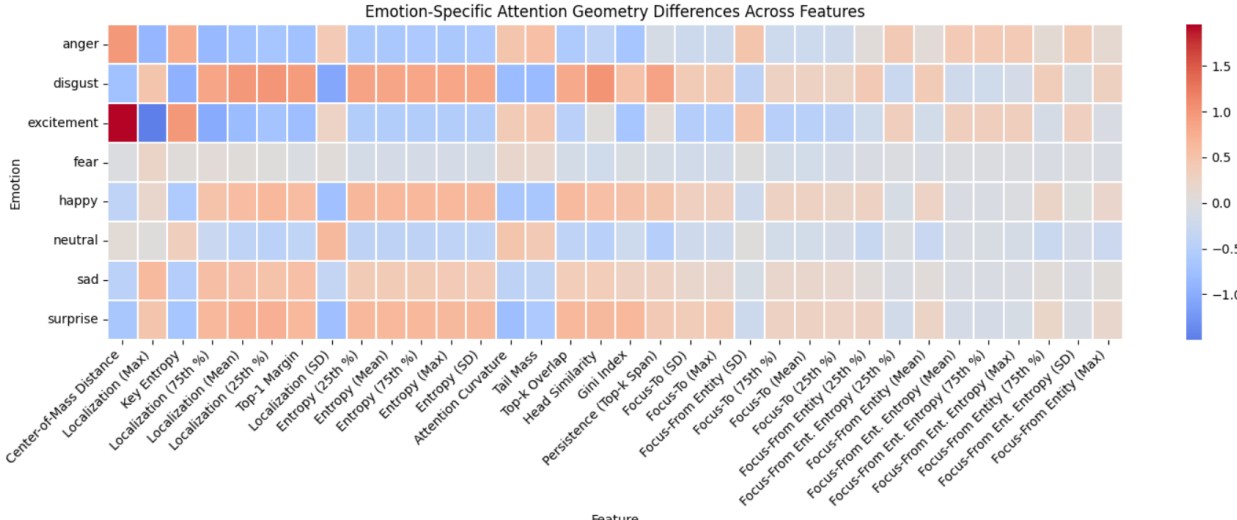

Figure 3: Emotion-specific differences in attention geometry across features. The heatmap reports one-vs-rest Cohen's d effect sizes, comparing each emotion to all others for each attention feature. Colors indicate the direction and magnitude of deviation relative to the global distribution. Features are ordered by across-emotion variance, highlighting attention dimensions most sensitive to emotional tone. Sarcasm is omitted for clarity due to its extreme divergence.

Moderate predictive power was also observed for focus-to features (AUC ≈ 0.68), measuring attention directed toward the answer span, and for entropy-based measures from the same distributions (AUC ≈ 0.64–0.65), where lower entropy corresponded to higher accuracy. Spatial geometry metrics such as center-of-mass distance and locality quantiles showed smaller but consistent effects, implying that moderate spatial spread—neither overly local nor diffuse—supports performance. Together, these findings suggest that bidirectional attentional flow between answer and context is a key correlate of accuracy, and that emotional tone, by modulating this flow, may in turn influence reasoning quality.

## 3.2 Attention Varies with Emotions

Having shown that attention geometry contains substantial information about question-answering accuracy, the next step is to examine how attention structure varies with emotional expression. Using the same per-example aggregated attention feature vectors described above, we trained a multi-class random forest classifier to predict the emotion label of each passage from attention-derived features alone. Random forests were used to capture non-linear relationships between attention features and emotion while remaining robust to correlated inputs. Performance was evaluated using 5-fold stratified cross-validation and reported using macro-averaged F1 and overall accuracy. This classifier achieved a macro-F1 of 0.75 and an accuracy of 86%, indicating that emotional tone leaves a measurable imprint on the model's internal attention geometry.

Beyond predictability, we examined how individual attention features vary across emotions by comparing their average values and distributional summaries across emotion classes. Multiple features exhibited consistent emotion-dependent shifts, including the standard deviation and lower quantiles of locality, top-1 margin, key entropy, mean locality, center-of-mass distance, overall entropy, curvature, tail mass, persistence across layers, and the Gini coefficient of attention mass. These differences span spatial, statistical, and depth-wise properties of attention, indicating that affective tone modulates attention geometry at multiple levels of the network.

Emotion-specific differences in attention geometry are visualized in Figure 3 using a one-vs-rest effect-size matrix. For each attention feature and each emotion, we compute Cohen's d (Cohen, 1988) between samples labeled with that emotion and all remaining samples pooled together. This yields a signed, standardized measure of how strongly a given feature is amplified or suppressed for one emotion relative to the rest, independent of sample size.

The resulting matrix is visualized as a heatmap, where rows correspond to emotions and columns correspond to attention features aggregated at the example level. Positive values indicate that a feature tends to take larger values for a given emotion than for others, while negative values indicate relative suppression. Features are ordered by their variance across emotions, so that dimensions with the strongest emotion-dependent differences appear first. Sarcasm is excluded from the main panel because its attention geometry diverges sharply from all other emotions, saturating the color scale and obscuring finer-grained patterns (see Appendix C).

This visualization provides a compact summary of the multifeature "signature" associated with each emotion. Among the remaining emotions, center-of-mass distance shows the clearest separation: excitement exhibits the greatest spatial spread of attention, while surprise and sadness concentrate attention locally near salient tokens. This pattern is mirrored in locality and entropy measures—high-arousal emotions such as excitement and anger display diffuse, exploratory attention with broader spatial reach and higher entropy, whereas low-arousal or negative emotions such as sadness and disgust exhibit tightly focused, convergent attention with lower entropy. Even emotions that appear visually similar, such as sadness and surprise or excitement and anger, maintain distinct geometric configurations, indicating that each emotion leaves a characteristic imprint on the model's attentional landscape.

Figure 4 visualizes pairwise distances between attention maps for different emotions, computed from mean attention patterns across the final transformer layers. The figure reveals interpretable differences in how the model allocates attention under varying emotional tones.

Sarcasm exhibits one of the largest shifts relative to other emotions. As shown in Figure 6, its attention patterns are markedly less localized, indicating a broader and more diffuse focus across tokens. Fear likewise produces substantial shifts, though their spatial distribution is more irregular. This irregularity suggests that transitions to fear correspond to intermediate levels of entropy and localization, as illustrated in Figure 3. Shifting to excitement, by contrast, increases the relative weight assigned to distant tokens, reflecting a larger center-of-mass distance. Each emotion pair in Figure 4 thus exhibits a distinct geometric signature, implying that the model encodes and processes emotional tone through systematically different attentional configurations.

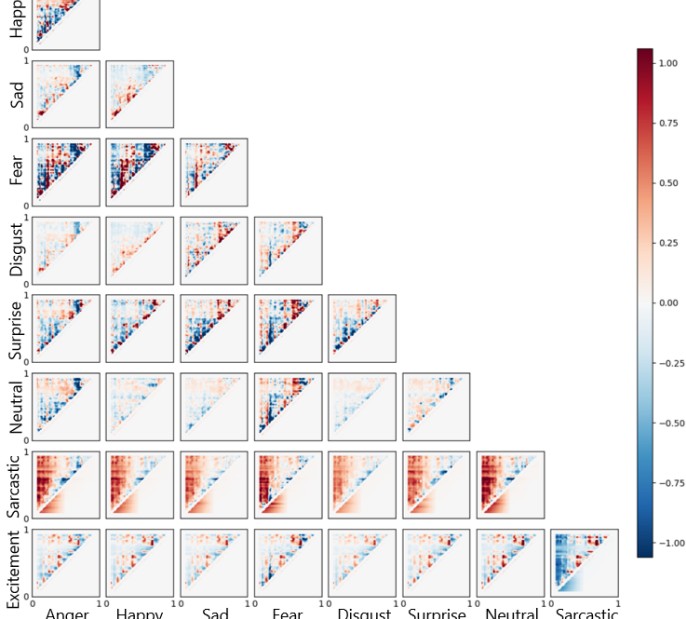

Figure 4: Differences in the attentional pattern between emotions. For each query token, attention differences across keys are standardized by subtracting the row mean and dividing by the row standard deviation, highlighting relative redistribution of attention rather than absolute magnitude changes.

## 3.3 Consequences

Overall, emotional tone is associated with systematic differences in attentional geometry. High-arousal emotions (e.g., excitement, anger) are characterized by more diffuse, exploratory attention with broader spatial spread, whereas low-arousal or negative emotions (e.g., sadness, disgust) exhibit more localized, convergent focus. Sarcasm displays a distinct pattern, combining wide spatial reach with sharp concentration on isolated tokens. These differences in attention structure are correlated with variation in question-answering performance, suggesting that emotional tone is associated with systematic differences in how information is attended to and organized during processing.

Taken together, these results indicate that emotional tone corresponds to consistent, measurable shifts in attention geometry that co-vary with comprehension performance. These findings are not intended to be interpreted prescriptively; rather, they reflect differences in how the model responds to and organizes information under different emotional expressions. Motivated by this observation, we introduce a dataset designed to study emotion-conditioned effects in LLM reasoning, along with a training framework that constrains emotion-conditioned representational variation during training.

# 4 Dataset Creation

A new dataset, Affect-Uniform ReAding QA (AURA-QA), is introduced to study how emotional tone influences question answering. Existing QA datasets are either synthetic, emotionally imbalanced, or lack sufficient contextual depth, motivating the need for an affect-aware corpus built from naturally written text. The design goals are threefold:

---

**Bloom Level 2 (Understand)**
**Passage excerpt:** "Raw meat ... does not appeal ... the idea of using it for food is disgusting."
**Question:** What do civilized persons find disgusting?
**Answer:** Raw meat

**Bloom Level 3 (Apply)**
**Passage excerpt:** "... the stony nature of the plains would soon disable an unshod horse ..."
**Question:** Why can't unshod horses chase guanacos?
**Answer:** Stony plains

---

Table 2: Representative examples of Bloom Level 2 (comprehension) and Level 3 (application) questions.

1. **Human authorship:** passages are drawn from naturally written text rather than machine-generated or crowd-sourced content.

2. **Emotional coherence:** each passage is dominated by a single primary emotion.

3. **Contextual adequacy:** passages contain sufficient length and narrative structure to support reasoning-based QA tasks.

Text is sourced from Project Gutenberg (pro), a large repository of public-domain books that span diverse genres and authorship styles. English-language narrative texts that were dominated by dialogue or poetry are filtered out. Preprocessing involved sentence segmentation, normalization, and removal of non-textual artifacts such as metadata and chapter headings.

## 4.1 Segment Construction

To construct emotionally coherent passages, a transformer-based emotion classifier fine-tuned on benchmark emotion datasets (Appendix A) is applied. The classifier outputs a probability distribution across $K = 9$ emotion categories corresponding to Ekman's six basic emotions (Ekman, 1992): joy, sadness, anger, fear, disgust, and surprise, supplemented with *neutral*, *sarcasm*, and *excitement*. This taxonomy is used operationally rather than as a claim about psychological primacy. In particular, *sarcasm* is included as a pragmatic affective mode that frequently induces reasoning failures in QA systems, rather than as a basic emotion. For each sentence $s_i$, its predicted emotion label and confidence margin are defined as:

$$e_i = \arg\max_k p_i^{(k)}, \qquad m_i = p_i^{(e_i)} - \max_{k \neq e_i} p_i^{(k)}.$$

A sentence is retained if $m_i \geq 0.25$, ensuring that the predicted emotion is sufficiently dominant over competing classes. A sweep over thresholds from 0.05 to 0.50 revealed a smooth trade-off between dataset scale and label confidence, with no sharp inflection point; 0.25 was selected as a balanced operating point (Appendix D.1).

Consecutive sentences that satisfy this margin criterion and share the same dominant emotion are aggregated into contiguous segments:

$$S_j = \{s_t, \ldots, s_{t+n}\} \text{ s.t. } e_{t:t+n} = e_t, \ m_{t:t+n} \geq 0.25$$

Each segment must contain at least three sentences *or* a minimum of forty words, and is capped at 150 words to maintain readability. Segments failing to meet these thresholds are discarded. This procedure yields passages that are locally consistent in affective tone while providing sufficient narrative context for downstream tasks.

**Segment Filtering.** To improve label reliability, a second-stage validation step is applied. For each candidate passage, three LLMs—LLaMA 3.3 70B (Grattafiori et al., 2024), Gemma 3–27B (Team et al., 2025), and Qwen 3 32B (Yang et al., 2025)—independently verify whether the target emotion is the dominant affective tone of the passage. A passage is retained only if all three models agree. The reliability of this procedure was assessed by comparing LLM consensus decisions to majority judgments from three human annotators. LLM–human agreement reached 58%, compared to 63% human–human agreement, indicating that LLM consensus operates within the range of observed annotator variability (Appendix D.2). Given the prohibitive cost of full human validation, LLM consensus is used as a scalable proxy, and emotion labels are treated as weakly supervised.

## 4.2 QA Construction

For each emotionally coherent passage, the same three LLMs were prompted to generate candidate question–answer pairs grounded in the passage. Prior work shows that multi-model generation reduces mode collapse and yields richer synthetic data (Reichman et al., 2025c). The models were instructed to produce questions within Bloom's Taxonomy (Bloom et al., 1964) levels 2 and 3. Level 1 questions were excluded for being too trivial, while levels 4–6 were omitted due to their reliance on subjective judgment. To illustrate the distinction between levels 2 and 3, Table 2 presents representative examples drawn from the dataset. Each model generated five candidate question–answer pairs using stochastic decoding at temperature $T = 1.0$. Questions were required to be answerable using only the passage, and answers were constrained to 1–3 words.

**QA Filtering.** To control question difficulty while preserving validity, a dual-model filtering procedure inspired by Samuel et al., 2024 is applied. For each candidate question, the originating LLM was required to correctly answer the question using the passage as context, confirming that the question was grounded and unambiguous. The same question was then evaluated using a smaller model from the same model family. A question was retained only if the larger model answered correctly while the smaller model failed. This criterion primarily modulates difficulty rather than question quality.

**Question–Answer Agreement** For each sampled QA pair, three human annotators independently evaluated *answerability*, *grounding*, and *non-triviality* using binary judgments. A question was considered *human-valid* if the majority of annotators marked it as answerable, grounded, and non-trivial. For additional experimental details on the human evaluations, see Appendix D.2.

Across both model-passed and model-failed subsets, human annotators judged the vast majority of questions to be valid (87.6%), indicating that the generation process produces generally well-formed and grounded questions. However, substantial differences emerge in answer difficulty: questions that pass the dual-model filter are markedly harder for humans to answer exactly (63.4%->42.8%), despite exhibiting comparable human-rated validity. This pattern demonstrates that the filtering procedure primarily modulates question difficulty rather than removing low-quality or ill-posed items. Per-emotion QA agreement statistics, including human validity rates and answer matching accuracy, are reported in Table 3.

**Dataset Statistics.** The dataset contains 14,400 QA pairs evenly distributed across nine emotions, with balanced Bloom-level splits and comparable model contributions. Table 4 summarizes the final dataset by emotion category. Passage lengths are broadly consistent due to uniform segmentation constraints, with neutral passages exhibiting greater length due to lower affective density. Question lengths remain short by design. Questions are evenly split between Bloom levels 2 and 3 for every emotion, ensuring controlled reasoning difficulty across affective conditions. Contributions from each generating model are balanced across emotions and reasoning levels.

# 5 Multi-Emotional Reading

The observed relationships between emotional tone, attention geometry, and question-answering performance suggest that emotional information is reflected in the model's internal representations. Building on recent efforts to model affective structure in representation learning (Reichman et al., 2025a), this work introduces an emotionally regularized training framework that integrates a navigable emotional latent space directly into the optimization process. By em-

| Emotion | Filtered Out | | | | | Retained | | | | |
|---|---|---|---|---|---|---|---|---|---|---|
| | Valid | Answerable | Grounded | Non-trivial | Answer Match | Valid | Answerable | Grounded | Non-trivial | Answer Match |
| Anger | 0.822 | 0.836 | 0.986 | 0.973 | 0.521 | 0.897 | 0.897 | 1.000 | 1.000 | 0.368 |
| Disgust | 0.880 | 0.893 | 1.000 | 0.987 | 0.627 | 0.958 | 0.972 | 1.000 | 0.986 | 0.408 |
| Excitement | 0.887 | 0.930 | 1.000 | 0.958 | 0.704 | 0.901 | 0.915 | 0.986 | 0.986 | 0.465 |
| Fear | 0.900 | 0.914 | 1.000 | 0.986 | 0.571 | 0.809 | 0.824 | 0.985 | 0.985 | 0.485 |
| Happy | 0.817 | 0.845 | 1.000 | 0.972 | 0.690 | 0.855 | 0.870 | 1.000 | 0.986 | 0.420 |
| Neutral | 0.738 | 0.831 | 0.985 | 0.862 | 0.585 | 0.873 | 0.909 | 1.000 | 0.964 | 0.473 |
| Sad | 0.958 | 0.958 | 1.000 | 1.000 | 0.653 | 0.851 | 0.866 | 0.985 | 0.985 | 0.448 |
| Sarcastic | 0.907 | 0.920 | 0.987 | 0.973 | 0.627 | 0.838 | 0.838 | 0.985 | 1.000 | 0.397 |
| Surprise | 0.959 | 0.959 | 1.000 | 1.000 | 0.726 | 0.900 | 0.914 | 1.000 | 0.986 | 0.386 |
| Avg. | 0.874 | 0.898 | 0.995 | 0.968 | **0.634** | 0.876 | 0.889 | 0.994 | 0.986 | **0.428** |

Table 3: Human evaluation of question–answer quality by emotion. **Valid** denotes majority agreement that a question is answerable, grounded, and non-trivial. **Answer Match** measures majority exact (token-set) match between the provided answer and human answers. Grounding rates are near ceiling across both subsets, indicating that the dual-model filter primarily modulates difficulty rather than question validity.

| Emotion | #Q | Passage Length | Question Length | Bloom L2 (%) | | | Bloom L3 (%) | | |
|---|---|---|---|---|---|---|---|---|---|
| | | (mean words) | (mean words) | Gemma | LLaMA | Qwen | Gemma | LLaMA | Qwen |
| Anger | 1600 | 48.5 | 7.9 | 31.4 | 35.9 | 32.8 | 31.0 | 34.2 | 34.8 |
| Disgust | 1600 | 44.3 | 8.1 | 30.2 | 37.9 | 31.9 | 27.1 | 36.5 | 36.4 |
| Excitement | 1600 | 59.4 | 7.6 | 29.6 | 35.9 | 34.5 | 27.5 | 36.2 | 36.2 |
| Fear | 1600 | 57.4 | 7.3 | 30.5 | 38.6 | 30.9 | 25.2 | 37.5 | 37.2 |
| Happy | 1600 | 77.3 | 7.8 | 30.5 | 38.1 | 31.4 | 26.1 | 37.0 | 36.9 |
| Neutral | 1600 | 176.8 | 8.4 | 21.9 | 43.9 | 34.2 | 22.6 | 39.5 | 37.9 |
| Sad | 1600 | 69.6 | 7.4 | 30.2 | 36.6 | 33.1 | 29.4 | 35.5 | 35.1 |
| Sarcastic | 1600 | 51.4 | 8.1 | 31.9 | 35.4 | 32.8 | 28.9 | 35.5 | 35.6 |
| Surprise | 1600 | 48.4 | 7.9 | 26.5 | 43.4 | 30.1 | 28.9 | 35.8 | 35.4 |
| **Average** | 1600 | 70.8 | 7.8 | 29.2 | 38.6 | 32.2 | 27.4 | 36.4 | 36.2 |

Table 4: Dataset statistics by emotion category. We report the number of questions, mean passage length, and mean question length (in words). For all emotions, questions are evenly split between Bloom level 2 and Bloom level 3 (50% each). Columns report the percentage of questions contributed by each LLM within each Bloom level.

bedding this latent structure within a novel emotional regularization loss, our method enhances question–answering performance across datasets exhibiting diverse emotional tones (Figure 5).

## 5.1 Method

Following prior work (Reichman et al., 2025a), emotional latent spaces are constructed from model activations via centered singular value decomposition applied to sentence-level representations. Concretely, for each example we run the model and collect activations from a chosen layer/module (we use both MLP hidden states and attention projection outputs in separate extractions). Token-level vectors are mean-pooled to form a single sentence embedding, and embeddings are stacked over a set of inputs to form a data matrix $X \in \mathbb{R}^{N \times d}$. The matrix is centered and decomposed via SVD, $X_C = U\Sigma V^\top$, with the top $k$ right-singular vectors defining an orthonormal basis for the emotional latent space. To ensure that affective variation dominates the decomposition, this procedure is performed on a synthetic parallel corpus (Reichman et al., 2025b) in which neutral sentences are rewritten into multiple emotions, making emotion the primary structured difference across samples. The resulting basis serves as the emotional latent space referenced throughout this paper and is used directly in the proposed regularization loss.

(Reichman et al., 2025a) analyzed the representations from the derived emotional latent spaces showing that the emotional latent spaces derived via centered SVD exhibit stable and meaningful structure across models, layers, and datasets. Emotional directions extracted from synthetic parallel data align closely with those recovered from human-written text across language styles and languages, with centroid cosine similarities typically in the $(0.8 - 0.9)$ range.

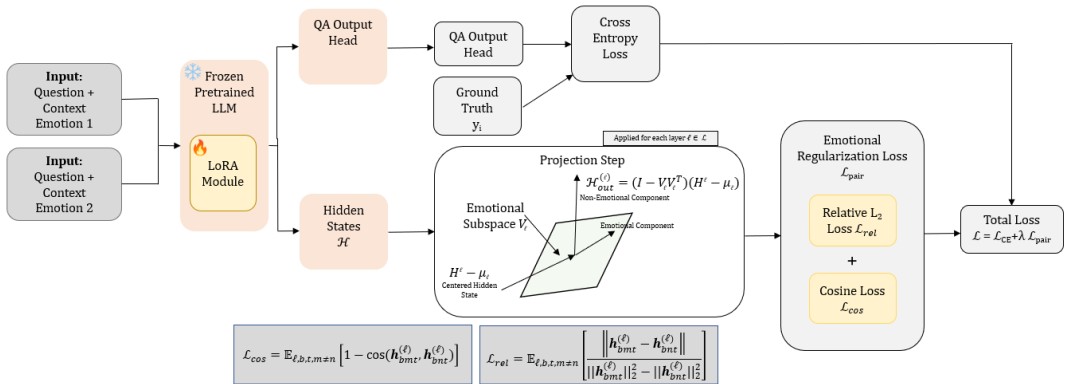

Figure 5: Overview of the proposed emotional regularization framework.

Global relational structure is largely preserved, as reflected by low stress values when comparing emotion geometries across datasets (generally $< 0.2$), while average distortion remains moderate for most layers, indicating limited local warping under projection. Mean-squared error between projected and reference emotion representations is correspondingly low, supporting faithful transfer of affective structure. Together, these complementary metrics indicate that the emotional latent space is directionally coherent and functionally meaningful, even though strict geometric isometry is not assumed.

Texts inherently vary in emotional valence, a property reflected in the model's emotional latent space. Nonetheless, instances arise in which affective tone inadvertently influences semantic interpretation, even among semantically equivalent inputs with differing emotional expressions. Within the proposed framework, **training objectives constrain emotional variability to the designated latent subspace**, thereby mitigating the propagation of affective representations beyond their intended domain.

Because the emotional latent space is defined independently at each layer, modifications confined to the emotional subspace of an upstream layer can inadvertently propagate into non-emotional representations downstream. The proposed method mitigates such cross-layer spillover by encouraging each layer's emotional processing to constrain its downstream effects within the corresponding emotional latent subspace, thereby preserving the separation between affective and non-affective representations throughout the network.

To this end, a Low-Rank Adaptation (LoRA) module (Hu et al., 2022) is optimized under a dual objective consisting of a standard question–answering (QA) loss and an emotional regularization loss:

$$\mathcal{L} = \mathcal{L}_{CE} + \lambda \mathcal{L}_{\text{pair}},$$

where $\mathcal{L}_{CE}$ is the cross-entropy loss for QA and $\mathcal{L}_{\text{pair}}$ enforces emotional consistency across variants of the same context.

Before computing the regularization loss, hidden states are mean-centered and projected onto the complement of each layer's emotional latent subspace. This isolates emotion-invariant representations by removing the components associated with emotional variation:

$$\mathcal{H}_{\text{out}} = (I - V_\ell V_\ell^\top)(\mathcal{H} - \mu_\ell),$$

where $V_\ell$ represents the basis vectors of the emotional subspace for layer $\ell$, and $\mu_\ell$ is the layer-wise mean activation. This projection ensures that the regularization loss operates only on the non-emotional components of the representations.

The emotional regularization loss is defined as:

$$\mathcal{L}_{\text{pair}} = \alpha \, \mathcal{L}_{\text{rel}} + \beta \, \mathcal{L}_{\text{cos}}.$$

Here, $\mathcal{L}_{\mathrm{rel}}$ regularizes relative $L_2$ differences between representations, making them scale-invariant and discouraging large norm disparities, while $\mathcal{L}_{\mathrm{cos}}$ regularizes angular differences, encouraging representations of emotional variants to remain directionally aligned.

$$\mathcal{L}_{\mathrm{rel}} = \frac{1}{|\mathcal{L}|} \sum_{\ell \in \mathcal{L}} \frac{1}{BMT} \sum_{b,t} \frac{1}{M^2} \sum_{m \neq n} \frac{\|h_{bmt}^{(\ell)} - h_{bnt}^{(\ell)}\|_2^2}{\|h_{bmt}^{(\ell)}\|_2^2 + \|h_{bnt}^{(\ell)}\|_2^2 + \varepsilon} \, c_{bt}$$

$$\mathcal{L}_{\mathrm{cos}} = \frac{1}{|\mathcal{L}|} \sum_{\ell \in \mathcal{L}} \frac{1}{BMT} \sum_{b,t} \frac{1}{M^2} \sum_{m \neq n} \left( 1 - \cos\left( h_{bmt}^{(\ell)}, h_{bnt}^{(\ell)} \right) \right) c_{bt}.$$

In these equations, $B$ is the batch size, $M$ is the number of emotions used per example, and $T$ is the number of tokens over which the loss is computed. $c_{bt} \in \{0, 1\}$ is a context mask ensuring that the loss is applied only to context tokens, excluding the system prompt or question tokens. $h_{bmt}^{(\ell)}$ denotes the hidden state at layer $\ell$, batch index $b$, emotion sample $m$, and token $t$.

The proposed regularization framework thus maintains a separation between affective and semantic representations, ensuring emotional nuance without compromising interpretive stability.

## 6  Experimental Setup

The proposed method is evaluated using three LLMs, LLaMA-3.1-8B, Ministral, and Olmov2, selected for their compatibility with the emotional latent space derivation procedure described in (Dubey et al., 2024; AI, 2024; OLMo et al., 2024; Reichman et al., 2025a).

LoRA modules were trained on several datasets, including NQ, TweetQA, FriendsQA, and AURA-QA (Kwiatkowski et al., 2019; Xiong et al., 2019; Yang & Choi, 2019). For each dataset, context passages were synthetically rewritten using an emotion translation model (Reichman et al., 2025c) to generate emotionally diverse yet semantically equivalent variants. As baselines, models were also trained on the original human-written datasets and on the multi-emotional variants without applying the emotional regularization loss. Each LoRA module was subsequently evaluated on TweetQA, FriendsQA, and the new dataset.

LoRA modules were affixed to every layer of the networks. Training was performed using the AdamW optimizer (Loshchilov & Hutter, 2019) with a learning rate of 3e-4 and an $\ell_2$ weight decay of 1e-2. A cosine learning rate scheduler with a 50-step warmup phase was employed.

Data sampling was random, except that each question appeared twice within a batch, once for each emotion in a randomly selected emotion pair. The batch size was determined dynamically based on the number of examples that filled 1,200 tokens. Training continued until convergence.

## 7  Results

Previous experiments in Table 1 showed that LoRA fine-tuning on emotional contexts improves QA performance by approximately 10% across emotions. Table 5 shows results when LoRA fine-tuning is extended with the proposed emotional regularization framework shown in Figure 5. The goal is to determine whether the regularization framework can improve reading comprehension in emotionally varied contexts while not harming emotionally non-varying contexts. Models are trained on a single dataset and evaluated both in-domain and out-of-domain. Three settings are reported: (i) LoRA fine-tuning without emotional augmentation or regularization, (ii) training with multi-emotion synthetic rewrites, and (iii) training with both augmentation and emotional regularization.

Under LoRA **training** on the fully neutral Natural Questions (NQ) dataset (the columns under "Natural Questions" in Table 5), adding emotion regularization to multi-emotion augmentation improves performance by an average of 3.03% across models and evaluation datasets. This indicates increased robustness to emotional variation despite the emotional homogeneity of the base training data. Multi-emotion augmentation alone yields a small mean decrease of 0.48%; however, performance is unchanged or improved in approximately 50% of model–dataset pairs, suggesting heterogeneous rather than uniformly negative effects.

| | **LoRA Fine-Tuning** | | | | | | | | | | | |
|---|---|---|---|---|---|---|---|---|---|---|---|---|
| | **Natural Questions** | | | **TweetQA** | | | **FriendsQA** | | | **AURA-QA** | | |
| **Test Set** | **Llama** | **Ministral** | **Olmov2** | **Llama** | **Ministral** | **Olmov2** | **Llama** | **Ministral** | **Olmov2** | **Llama** | **Ministral** | **Olmov2** |
| Natural Questions | 61.6% | 60.0% | **61.1%** | 56.7% | 56.5% | 56.1% | 58.3% | **59.9%** | 59.4% | 51.3% | 54.5% | 52.3% |
| TweetQA | 59.5% | 56.4% | 58.6% | 73.1% | **74.9%** | 72.7% | 69.0% | 68.4% | **70.1%** | 64.1% | 65.3% | 62.1% |
| FriendsQA | 46.4% | 46.1% | 51.1% | 46.4% | 48.6% | 46.8% | 54.4% | 38.3% | 67.3% | 37.1% | 40.6% | 37.1% |
| AURA-QA | 35.2% | 35.4% | 36.2% | 42.8% | **43.4%** | **43.5%** | 40.4% | **47.0%** | 44.1% | 49.9% | **51.7%** | **47.4%** |
| | **LoRA + Multi-Emotion Data Augmentation** | | | | | | | | | | | |
| | **Natural Questions** | | | **TweetQA** | | | **FriendsQA** | | | **AURA-QA** | | |
| **Test Set** | **Llama** | **Ministral** | **Olmov2** | **Llama** | **Ministral** | **Olmov2** | **Llama** | **Ministral** | **Olmov2** | **Llama** | **Ministral** | **Olmov2** |
| Natural Questions | 56.1% | 59.2% | 59.8% | 56.4% | 54.9% | 56.4% | 62.3% | 57.0% | **62.4%** | 51.5% | **57.3%** | 51.7% |
| TweetQA | 57.2% | 62.5% | 61.9% | 72.2% | 71.6% | 71.3% | 67.6% | 68.8% | 68.5% | 59.5% | **69.5%** | 62.0% |
| FriendsQA | 39.6% | 47.2% | 49.2% | 44.4% | 46.4% | 45.2% | **62.2%** | 51.4% | 65.1% | 34.8% | **46.4%** | 38.1% |
| AURA-QA | 36.1% | 36.8% | 36.2% | 37.9% | 38.8% | 41.3% | 40.6% | 37.6% | 40.8% | 36.2% | 44.2% | 39.2% |
| | **LoRA + Multi-Emotion Data Augmentation + Emotion Regularization** | | | | | | | | | | | |
| | **Natural Questions** | | | **TweetQA** | | | **FriendsQA** | | | **AURA-QA** | | |
| **Test Set** | **Llama** | **Ministral** | **Olmov2** | **Llama** | **Ministral** | **Olmov2** | **Llama** | **Ministral** | **Olmov2** | **Llama** | **Ministral** | **Olmov2** |
| Natural Questions | **62.7%** | **61.0%** | 59.8% | **58.6%** | **58.8%** | 56.7% | **63.7%** | 59.6% | 61.8% | **58.8%** | 57.0% | **58.3%** |
| TweetQA | **64.0%** | **65.4%** | **62.7%** | **74.5%** | 72.9% | **74.3%** | **71.1%** | **69.0%** | 69.9% | **67.2%** | 68.1% | **68.5%** |
| FriendsQA | **49.6%** | **49.7%** | **55.2%** | **49.9%** | **50.2%** | 48.3% | **62.2%** | **61.1%** | **68.8%** | **42.2%** | 44.2% | **53.0%** |
| AURA-QA | **37.7%** | **39.0%** | **36.8%** | **44.0%** | 43.0% | 41.2% | **41.4%** | 41.2% | 42.2% | 44.6% | 43.5% | 42.6% |
| Avg Δ | 2.83% | 4.27% | 1.91% | 2.00% | 0.35% | 0.35% | 4.06% | 4.31% | 0.44% | 2.62% | 0.19% | 6.78% |

Table 5: LoRA training and emotional regularization results when trained and tested across multiple datasets.

When **evaluating** on NQ (the rows of results for "Natural Questions"), models trained with emotion regularization consistently outperform their non-regularized counterparts across architectures and training conditions. This suggests that constraining affective variation to a designated latent subspace during training can improve generalization to emotionally neutral datasets.

When training on TweetQA and FriendsQA, which already contain emotional variation, multi-emotion augmentation yields little benefit, improving performance in only two isolated cases. In contrast, emotional regularization yields average absolute improvements of 0.9% in-domain and 2.9% out-of-domain, indicating larger gains under distribution shift. This indicates that exposure to emotional variation alone is insufficient to induce emotion–semantic disentanglement, whereas explicit regularization is effective.

Finally, on the newly constructed AURA-QA dataset, emotional regularization consistently improves out-of-domain performance, though the source of gains varies by model. For Ministral, the largest improvements arise from multi-emotion augmentation alone, with additional gains from regularization. For Olmov2, both augmentation and emotional regularization improve performance, with the latter yielding larger gains. For LLaMA, only emotional regularization improves out-of-domain performance. Despite persistent emotion-specific accuracy differences on AURA-QA, the proposed regularization yields limited in-domain gains. The observed out-of-domain improvements indicate that emotion-conditioned representational drift remains a meaningful and correctable factor, while the in-domain results suggest that this drift interacts with additional sources of reasoning difficulty rather than acting in isolation. Because AURA-QA is intentionally balanced across emotions, it reduces the emotional distribution shift that motivates the regularizer. Under these conditions, enforcing emotion-invariant representations may suppress information that remains useful for the source task, producing a robustness-specialization tradeoff.

Appendix F contains an ablation study examining the effect of emotion-pair regularization.

# 8 Conclusion

Prior work has largely treated emotion either as a prediction target (e.g., sentiment or emotion classification) or as a dimension of model competence, such as measuring the emotional intelligence of LLMs. In contrast, this paper frames emotion as a latent factor associated with systematic variation in internal processing and downstream reasoning behavior in LLMs. Building on evidence that emotional information is organized within a shared latent space, it

is shown that emotional tone induces structured changes in attention geometry that correlate with question–answering performance. To enable controlled study of these effects, AURA-QA is introduced, an emotionally balanced question–answering dataset derived from human-authored texts, addressing limitations of prior synthetic or emotionally imbalanced benchmarks. Finally, we propose an emotional regularization framework that leverages this latent structure to constrain emotion-conditioned representational drift during training. Across datasets and models, this approach improves reading comprehension in both emotionally variable and emotionally non-varying datasets, with particularly strong gains under distribution shift and improvements on several in-domain benchmarks.

**Limitations:** While AURA-QA is derived from human-authored texts, several components of the dataset construction pipeline, including emotion verification and question–answer generation, rely on LLMs. Although model-specific biases are mitigated through multi-model agreement and validation procedures, residual artifacts or biases introduced through LLM-mediated annotation and generation may remain. In addition, the proposed emotional regularization framework targets a specific mechanism, emotion-conditioned non-emotional representational drift, it is not intended to address all sources of error in question answering. As a result, the benefits can be dataset- and setting-dependent, with particularly strong gains under distribution shift and more variable effects across in-domain settings. Finally, our analysis focuses primarily on attention geometry and does not exhaustively characterize all representational pathways through which emotion may influence reasoning. Moreover, the attention analyses presented here remain correlational in nature. While attention geometry is associated with both emotional tone and question-answering performance, the present study does not establish a causal relationship between these factors, leaving intervention-based analyses and broader representational investigations as important directions for future work.

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

# A  Emotion Classifier

In Section 2, the TweetQA, and FriendsQA datasets were introduced as human-written reading comprehension datasets containing emotional content. However, the context passages in these datasets do not include emotion labels. To address this, an emotion classification system was developed to generate silver-label annotations for the context passages.

The emotion classification system was trained on a combination of five datasets, all previously described in the main text: Go-Emotions Demszky et al. (2020), SemEval-2007 Task 14 Strapparava & Mihalcea (2007), CARER Saravia et al. (2018), EmoEvent-en Plaza-del-Arco et al. (2020), and Sarc Khodak et al. (2018). Combining these datasets allows the models to observe emotional expressions across a range of contexts, including tweets, news headlines, and Reddit posts, improving generalization. The final training set includes 323,776 labeled examples.

Four models were trained for emotion classification: T5-small, T5-large, ModernBERT-base, and ModernBERT-large Raffel et al. (2020); Warner et al. (2024). To preserve the pretrained representations, the models were frozen except for their last three transformer layers and the classification head. The last three layers were fine-tuned with a learning rate of 5e-4, while the classification head used a learning rate of 1e-3. Training used a weighted cross-entropy loss, where each class was weighted by the square root of the inverse of its frequency in the dataset. Because sarcasm was overrepresented in the dataset mixture, its subset was downsampled after the first epoch: only 20% of sarcasm examples were retained, and of those, 70% were high-entropy (difficult), 20% medium, and 10% easy. This resampling strategy ensured that sarcastic examples continued to appear during training, while preventing class imbalance and mitigating catastrophic forgetting of easier examples.

Each model's F1 score and accuracy are reported in Table 6. The average F1 scores ranged from 0.75 to 0.77, with some variation across emotions. Sarcasm and happiness were typically the best-performing classes, while disgust was the most challenging. Cases where the F1 score was high but the accuracy was relatively lower indicate a higher number of false positives, suggesting the models prioritized recall over precision for those classes. Models trained and evaluated on sarcasm achieved higher overall accuracy due to the large number of sarcastic examples in the dataset. However, when sarcasm was excluded from training, the F1 scores for the remaining classes remained stable, while overall accuracy dropped. This indicates that the sarcasm examples primarily affected the class distribution, not the classifier's ability to recognize other emotions.

The final system used a cascading classifier setup. In the first round, models trained with sarcasm included in the label set were used to predict whether the input was sarcastic. In the second round, models trained without sarcasm in the label set classified the specific emotion. If the first round detected sarcasm, the ensemble output was set to sarcastic. Otherwise, the system combined predictions all eight models to determine the final emotion label. The results of this setup are shown in Table 6. The ensemble achieved a +0.05 improvement in macro F1 and a +0.01 improvement in accuracy compared to individual models. This confirms that the ensembling method improves the quality of the generated emotion labels, providing reliable silver-label annotations and intent tags for downstream use.

| Model | Anger | Happy | Sad | Fear | Disgust | Surprise | Neutral | Sarcastic | Excitement | Average |
|-------|-------|-------|-----|------|---------|----------|---------|-----------|------------|---------|
| **With Sarcasm** | | | | | | | | | | |
| ModernBERT-base | 0.75/0.78 | 0.89/0.89 | 0.85/0.85 | 0.76/0.87 | 0.61/0.83 | 0.65/0.79 | 0.84/0.80 | 1.00/0.99 | 0.59/0.88 | 0.77/0.96 |
| ModernBERT-large | 0.71/0.77 | 0.89/0.91 | 0.82/0.85 | 0.77/0.89 | 0.60/0.81 | 0.60/0.67 | 0.80/0.70 | 1.00/1.00 | 0.54/**1.00** | 0.75/0.96 |
| T5-base | 0.66/0.70 | 0.86/0.93 | 0.80/0.82 | 0.76/**0.91** | 0.61/0.85 | 0.63/0.70 | 0.79/0.73 | 0.99/0.99 | 0.60/0.82 | 0.75/0.96 |
| T5-large | 0.68/0.77 | 0.88/0.92 | 0.80/0.82 | 0.78/0.86 | 0.62/0.74 | 0.64/0.65 | 0.80/0.72 | 1.00/0.99 | 0.59/0.86 | 0.75/0.96 |
| **Without Sarcasm** | | | | | | | | | | |
| ModernBERT-base | 0.75/0.77 | 0.90/0.90 | 0.85/0.90 | 0.78/0.84 | 0.60/0.71 | 0.59/**0.83** | 0.86/0.78 | - | 0.56/0.61 | 0.74/0.82 |
| ModernBERT-large | 0.75/0.74 | 0.90/0.90 | 0.85/0.81 | 0.80/0.87 | 0.61/**0.88** | 0.65/0.79 | 0.87/**0.83** | - | 0.64/0.70 | 0.76/0.83 |
| T5-base | 0.69/0.73 | 0.91/0.93 | 0.83/0.83 | 0.79/0.89 | 0.63/0.81 | 0.63/0.76 | 0.83/0.76 | - | 0.69/0.69 | 0.75/0.80 |
| T5-large | 0.72/0.77 | 0.91/0.93 | 0.84/0.83 | **0.82**/0.87 | 0.64/0.82 | 0.71/0.75 | 0.86/0.79 | - | 0.68/0.64 | 0.77/0.82 |
| **Ensemble** | | | | | | | | | | |
| Ensemble | **0.76/0.79** | **0.92/0.94** | **0.87/0.87** | **0.82**/0.89 | **0.66**/0.86 | **0.72**/0.78 | **0.87**/0.82 | **1.00/1.00** | **0.74**/0.81 | **0.82/0.97** |

Table 6: Emotion classification results per model. Each cell shows F1 / accuracy.

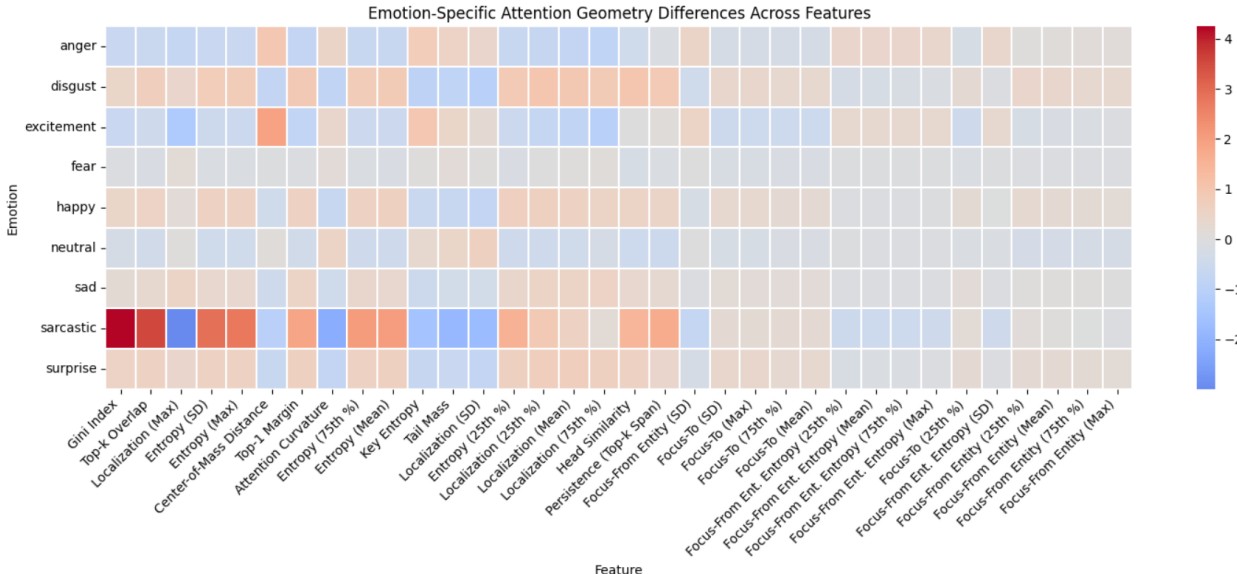

Figure 6: Differences in attention features by emotion. Sarcasm is included.

## B    Web Corpus Emotion Distribution

All statistics in Figure 1 are computed over a randomly sampled 10-million-document subset drawn from the Refined-Web dataset (Penedo et al., 2023), a large web-text corpus. Each document is labeled using the same emotion classifier described in Appendix A. To handle the substantial length variability of web documents, we do not truncate inputs. Instead, each document is tokenized and segmented into overlapping windows of up to 1024 tokens, with a stride of 64 tokens between consecutive windows. Emotion predictions are produced independently for each window, and a single document-level label is obtained by selecting the window whose prediction has the highest confidence. This procedure ensures that emotional content appearing anywhere in a document can influence the final label.

## C    Attention Features of Sarcasm

Sarcasm displays a uniquely skewed attention geometry as shown in 6. It yields a markedly higher Gini index, indicating that attention mass is distributed highly unevenly, with a few tokens receiving disproportionate focus. However, the localization (max) value is lower than for other emotions, implying that this concentration is not spatially compact, attention spikes occur across dispersed parts of the sequence rather than within a single contiguous region. Sarcasm also exhibits higher top-k overlap, meaning that multiple heads redundantly attend to these same sparse focal points, and higher entropy, suggesting increased variability or instability in token-level weighting. Combined with its low center-of-mass distance, these features indicate an attention pattern that is simultaneously redundant and fragmented: strongly peaked but spatially disjoint. This geometry aligns with the pragmatic duality of sarcastic language, anchored to key tokens that signal irony while diffusing elsewhere to preserve literal surface meaning.

## D    Dataset Details

### D.1    Margin Threshold Sensitivity

The sensitivity of passage segmentation to the margin threshold was evaluated by sweeping values from 0.05 to 0.50 in increments of 0.025. For each threshold, we measure (i) the average number of retained segments per emotion and (ii) the mean passage-level confidence, computed from the aggregated sentence-level emotion logits.

As shown in Figure 7, increasing the margin threshold induces a smooth and monotonic trade-off between dataset scale and label confidence. Lower margins retain substantially more segments but exhibit reduced confidence, while

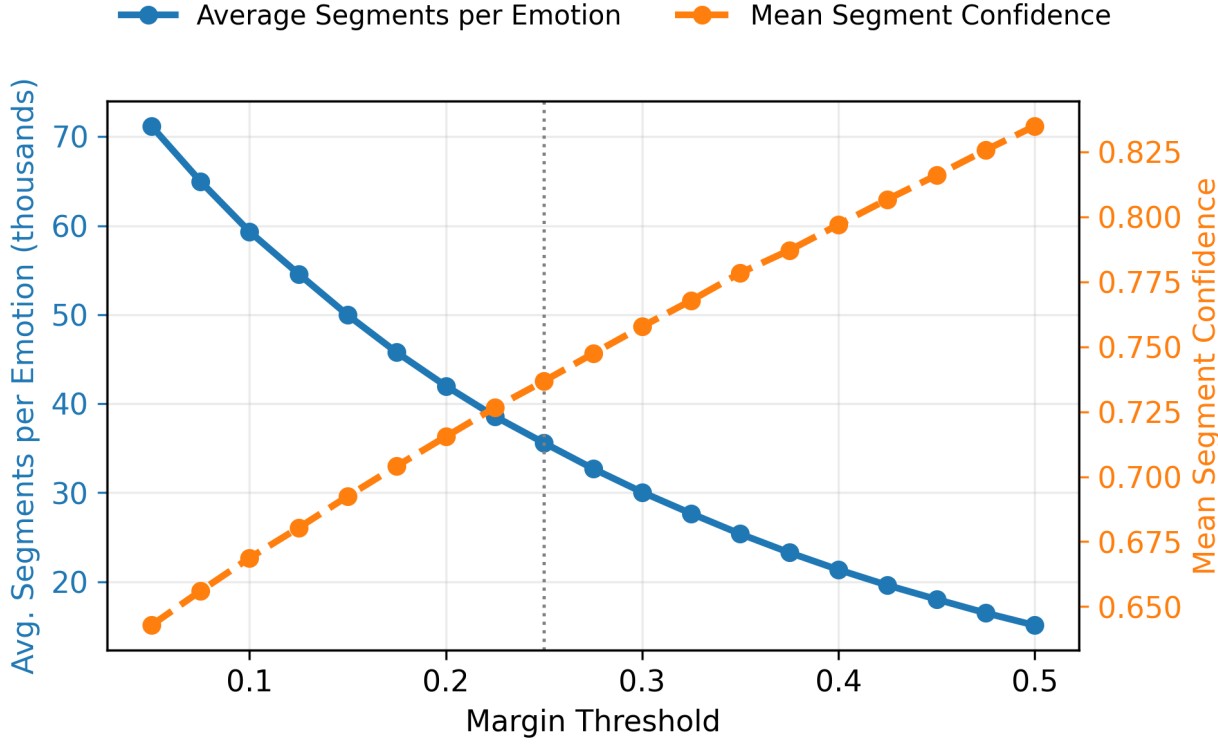

Figure 7: Sensitivity of passage segmentation to the emotion margin threshold.

stricter thresholds lead to sharp reductions in data volume with diminishing gains in confidence. No abrupt inflection point is observed.

Based on this analysis, a margin threshold of 0.25 was selected as a balanced operating point that preserves dataset scale while substantially improving label confidence.

## D.2 Agreement

**Sampling protocol.** To evaluate agreement under both confident and ambiguous conditions, controlled agreement studies were conducted for passage-level emotion verification and question–answer (QA) validation using a shared sampling strategy. For each emotion category (including neutral), we sampled 150 items balanced between cases accepted and rejected by LLM-based filtering (75 positive / 75 negative). This resulted in 1,350 passages for emotion agreement and a matched subset of QA pairs for question agreement.

**Agreement measures.** All analyses were performed at the item level using binary decisions. Each item was evaluated by three human annotators and three large language models (LLaMA 3.3 70B, Gemma 3–27B, and Qwen 3 32B). Three complementary measures were reported: (i) **Human–Human** agreement, defined as unanimous agreement among the three human annotators; (ii) **LLM–LLM** agreement, defined analogously as unanimous agreement among the three models; and (iii) **LLM–Human** agreement, defined as agreement between majority votes of humans and LLMs. Majority voting mitigates individual annotator noise while preserving genuine ambiguity.

**Passage-Level Emotion Agreement** Table 7 reports agreement statistics for passage-level emotion verification. Human–Human unanimity averages 62.6%, reflecting the inherent subjectivity of affective labeling in narrative text. LLM–LLM unanimity is higher at 85.2%, indicating strong internal consistency among models. Agreement between LLM majority votes and human majority votes reaches 58.1%, falling slightly below Human–Human agreement.

| Emotion | LLM–Human | Human–Human | LLM–LLM |
|---------|-----------|-------------|---------|
| Joy | 0.693 | 0.687 | 0.900 |
| Sadness | 0.533 | 0.807 | 0.787 |
| Anger | 0.587 | 0.713 | 0.860 |
| Fear | 0.553 | 0.727 | 0.700 |
| Disgust | 0.540 | 0.767 | 0.853 |
| Surprise | 0.573 | 0.660 | 0.867 |
| Neutral | 0.647 | 0.473 | 0.840 |
| Sarcasm | 0.573 | 0.247 | 0.940 |
| **Mean** | **0.581** | **0.626** | **0.852** |

Table 7: Inter-rater agreement for passage-level emotion verification. Human–Human and LLM–LLM denote unanimous agreement (3/3). LLM–Human compares majority votes (three LLMs vs. three humans).

Restricting evaluation to passages for which all three LLMs agreed increases LLM–Human agreement to 65.8%. This improvement indicates that LLM unanimity provides a useful confidence signal, identifying passages on which model judgments are more likely to align with human consensus.

| Criterion | Human–Human Unanimity |
|-----------|----------------------|
| Answerable | 0.71 |
| Grounded | 0.93 |
| Non-trivial | 0.79 |

Table 8: Human–Human agreement for QA quality criteria.

## Passage

I steamed up a bit, then swung down stream, and two thousand eyes followed the evolutions of the splashing, thumping, fierce river-demon beating the water with its terrible tail and breathing black smoke into the air.

**Does this passage express fear?**

○ Yes    ○ No

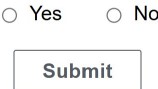

Figure 8: Human annotation interface for passage-level emotion verification. Annotators judged whether the target emotion was expressed as the dominant affective tone of the passage using a binary decision.

**Human Annotation Interfaces**  We employed two distinct Amazon Mechanical Turk (AMT) interfaces to support human validation during dataset construction: one for passage-level emotion verification and one for question–answer validation. Both interfaces were designed to minimize ambiguity and ensure consistent judgments across annotators.

Figure 8 shows the interface used for human verification of passage-level emotion labels. Annotators were presented with a passage and a target emotion label, and were asked to determine whether the target emotion was expressed as the *dominant affective tone* of the passage. Judgments were binary (Yes / No).

Figure 9 shows the interface used for validating generated question–answer pairs. Annotators were presented with a passage and an associated question, and were asked to evaluate the question along three dimensions:

- **Answerability:** Can the question be answered unambiguously using the passage alone?

**Passage**

The thought that he had been made by this man—made in
the semblance of a human being, yet denied by the manner
of his creation a place among the lowest of Nature's creatures
—filled him with fury, but it was not this thought that drove him
to the verge of madness.

**Question**

What filled him with fury?

**1. Answer the question using ONLY the provided passage.
(Your answer must be short: 1–3 words)**

answer…

**2. Can this question be answered unambiguously?**
○ Yes    ○ No

**3. Is the question clearly grounded in the passage?**
○ Yes    ○ No

**4. Is the question non-trivial?**
○ Yes    ○ No

Submit

Figure 9: Human annotation interface for question–answer validation. Annotators evaluated answerability, grounding, and non-triviality of questions with respect to the passage.

- **Grounding:** Is the question clearly grounded in the content of the passage, without requiring outside knowledge?

- **Non-triviality:** Is the question non-trivial, i.e., not answerable by copying a single phrase from the passage?

**Annotators.**   All annotations were performed by Amazon Mechanical Turk Master annotators with a minimum prior approval rate of 95%. Annotators were restricted to English-speaking countries to ensure reliable comprehension. Annotators were compensated at rates meeting or exceeding the U.S. federal minimum wage based on conservative task-time estimates. All annotations involved non-sensitive textual content, and no personally identifiable information was collected.

# E   LLM Prompts

**Passage-Level Emotion Verification Prompt**   Each candidate passage is verified by each LLM independently using the prompt below.

**System:** You are an emotion classifier.
Determine whether the following passage primarily expresses the emotion: `[EMOTION_LABEL]`
If it DOES express this emotion, reply EXACTLY with: `<answer>yes</answer>`
If it DOES NOT express this emotion, reply EXACTLY with: `<answer>no</answer>`
Do not include any explanation, reasoning, or additional text.
Passage: `[PASSAGE_TEXT]`

**Question–Answer Generation Prompt**   For each passage, each LLM generates multiple candidate QA pairs using the prompt below.

Figure 10: Ablation across emotion pairs.

---

**System:** You are a QA generation assistant.
Goal: Generate a question and a SHORT answer based on the passage.
RULES: 1. Output valid JSON. 2. The answer must be SHORT (1 to 3 words). 3. The answer must be factually supported by the passage.

---

The user prompt conditions on Bloom level:

---

**User:** `[BLOOM_INSTRUCTION]`
Passage: `[PASSAGE_TEXT]`

---

where:

---

**Bloom Level 2**: Ask a question checking comprehension of meaning or reason.
**Bloom Level 3**: Ask a question requiring using a fact or logic from the passage.

---

**Shared QA Answering Prompt (Self-Check and Small-Model Check)**  Both the self-check (same model re-answering its own question) and the small-model check (use of a smaller model from the same family) use an identical answering prompt.

---

**System:** You are a precise question-answering assistant.
You will be given a passage and a question. Answer using ONLY information from the passage.
Your answer MUST be 1–3 words taken exactly from the passage.
Return ONLY: { "answer": "..." }

---

# F  Which Emotion Best Regularizes Emotional Expression?

The method devised has a few components, each of which can be what improves model performance. This subsection breaks down the method into its components and tries to understand what drives the performance gains. Due to the computational cost of running ablation experiments, these experiments were primarily performed with Llama-3.1-8B.

Part of the algorithm is regularizing across emotions. To test the effect of which emotion pairs gives better performance, the model was trained with only one pair at a time. Figure 10 shows the results. There is a 7%-13% variation in performance depending on the dataset. For TweetQA, regularizing across a single emotion pair can potentially improve performance beyond the human-written baseline. The "best" emotion pair to regularize across, however, is not consistent across datasets. With each dataset showing different emotion pairs to work better for them.

To investigate why certain emotion pairs yield stronger regularization effects, we examine their geometric alignment in latent space. For each emotion pair, we compute the difference between their centroid representations within the emotional latent space and measure the cosine similarity of this direction to the corresponding pair in the multi-emotion NQ dataset. This quantity captures how consistently a given emotional contrast (e.g., anger–fear) is represented

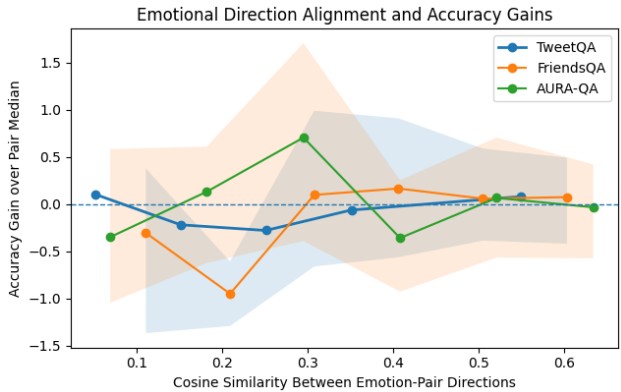

Figure 11: Relationship between emotion-pairs used for regularization and performance.

across datasets. We then compare this alignment to the z-scored accuracy gains obtained when using each pair for regularization (Figure 11).

For FriendsQA, performance increases approximately monotonically with alignment: emotion pairs whose directions closely match NQ's latent geometry yield the strongest gains. AURA-QA exhibits a mildly non-monotonic but structured response, with improvements at low alignment, a dip at intermediate alignment, followed by partial recovery at higher alignment before saturating. This suggests that while AURA-QA's emotional geometry is broadly compatible with the NQ manifold, regularization benefits depend on the alignment regime rather than scaling smoothly with consistency. In contrast, TweetQA shows a more sharply non-monotonic pattern, with strongest gains at weak alignment and degradation thereafter, consistent with its short, irony-heavy style. This likely reflects TweetQA's short, irony-heavy style, where emotional structure is poorly aligned with NQ's latent geometry. In such regimes, the regularizer acts as a corrective force when affective directions diverge strongly, but saturates or destabilizes once partial alignment emerges. Overall, when a dataset's emotional axes align with those of the training manifold, the regularizer can reliably suppress non-emotional variance while preserving semantics; when they diverge, gains are still possible, but become less predictable and more pair-dependent.

