# OpenReview forum: "Emotion is Not Just a Label: Latent Emotional Factors in LLM Processing"
_TMLR — Decision pending for TMLR_

### Review · Reviewer_w9o3 · 2026-04-06

**Summary Of Contributions:**

This paper studies emotion not as a prediction target, but as a latent factor that influences how large language models process and reason over text. It shows that emotional tone systematically affects question-answering performance, with significant variation across emotions even when the task itself is non-emotional. To explain this, the authors analyze attention geometry in transformer models and demonstrate that attention patterns (e.g., locality, entropy, and spatial spread) vary consistently with emotion and correlate with accuracy.

To enable controlled evaluation, the paper introduces AURA-QA, a new question-answering dataset with balanced emotional distributions constructed from human-authored text. Finally, the authors propose an emotional regularization framework that constrains emotion-induced representational drift during training by separating emotional and non-emotional components in latent space. Experiments across multiple datasets and models show that this method improves robustness, particularly under distribution shift.

Strengths include a novel conceptual framing, thorough empirical analysis of attention, and a well-motivated dataset and method. Weaknesses include reliance on LLM-generated data and limited causal evidence for the role of attention in driving performance differences.

**Audience:**

Yes

**Audience Explanation:**

Yes, the paper addresses an important and timely question about how non-semantic factors, such as emotional tone, influence the internal processing and reasoning behavior of large language models. This is highly relevant to the TMLR audience, particularly researchers working on interpretability, robustness, and representation learning in LLMs.

**Claims And Evidence:**

Yes

**Claims Explanation:**

The paper provides strong empirical evidence supporting its main claims. It demonstrates consistent performance differences across emotional contexts using both existing datasets and the newly introduced AURA-QA benchmark. The attention geometry analysis is thorough and shows that attention-derived features can predict both model accuracy (AUC ~0.75) and emotional labels, indicating a meaningful relationship between emotion and internal model behavior.

The proposed emotional regularization method is evaluated across multiple models and datasets, and results show consistent improvements, especially under distribution shift. Ablations further support the contribution of the regularization component.

However, while the evidence is convincing at a correlational level, the paper does not establish a clear causal link between attention changes and performance differences. Additionally, the dataset construction pipeline relies heavily on LLM-generated annotations and filtering, which introduces potential bias. Despite these limitations, the overall empirical support is strong and clearly presented.

**Requested Changes:**

1. The paper should provide stronger evidence for causality between attention geometry and performance. Currently, the analysis is correlational; controlled interventions (e.g., modifying attention patterns or directly testing the effect of attention constraints) would significantly strengthen the claims.

2. The reliance on LLMs for dataset construction (emotion labeling and QA generation) should be more critically discussed. In particular, the relatively low agreement between LLMs and human annotators suggests potential noise or bias in the dataset that could affect conclusions.

3. The evaluation is limited to question-answering tasks. Extending experiments to other reasoning tasks (e.g., multi-hop reasoning, math, or general benchmarks like MMLU) would strengthen the generality of the claims.

---

> ### Author Response · Authors · 2026-06-07
>
> We thank the reviewer for the positive assessment of the paper and for recognizing the value of the empirical analysis, the AURA-QA benchmark, and the emotional regularization framework.
>
> **Attention geometry and causality.** We agree that the attention-geometry analysis is correlational rather than causal. Our intention was to characterize systematic relationships between emotional tone, attention structure, and QA performance rather than establish a direct causal mechanism. Throughout Section 3, we intentionally use correlational language such as "associated with," "correlated with," and "co-vary with." We agree that intervention-based studies, such as directly modifying attention patterns or imposing attention constraints, would provide stronger evidence regarding causality. We clarified this distinction further in the revised manuscript and highlight such interventions as an important direction for future work.
>
> **LLM-mediated dataset construction.** We agree that the use of LLMs for emotion verification and QA generation introduces the possibility of synthetic artifacts and annotation noise. At the same time, the underlying passages are human-authored and sourced from Project Gutenberg, and we employ a multi-model consensus procedure (LLaMA, Gemma, and Qwen) together with human validation to reduce model-specific biases. The resulting dataset achieves 87.6% human validity for generated QA pairs, and LLM-human agreement improves substantially when restricted to unanimous LLM decisions. Our goal was therefore not to eliminate all possible generation artifacts, but rather to construct a scalable benchmark with strong quality controls. We agree that these limitations deserve additional discussion and have expanded the limitations section accordingly.
>
> **Generality beyond question answering.** We agree that evaluating emotion-conditioned effects beyond QA would be valuable. Our focus in this work was intentionally restricted to reading comprehension and question answering in order to isolate the phenomenon in a controlled setting. Extending the analysis to broader reasoning tasks, including multi-hop reasoning, mathematical reasoning, and general-purpose benchmarks such as MMLU, represents an important direction for future work and would help determine the extent to which the observed effects generalize beyond QA.

---

> > ### Comment · Reviewer_w9o3 · 2026-06-13
> > **response to rebuttal**
> >
> > Thank you for the detailed response.
> >
> > The clarification regarding attention geometry is helpful. I appreciate that the authors explicitly position the analysis as correlational rather than causal and have revised the manuscript to emphasize this distinction. Given this clarification, my concern is substantially addressed, though I still view intervention-based studies as an important future direction for establishing stronger mechanistic evidence.
> >
> > The additional discussion of the dataset construction process is also appreciated. The use of human-authored source passages, multi-model consensus filtering, and human validation strengthens confidence in the benchmark. However, I still believe the reliance on LLM-mediated labeling and QA generation remains a meaningful limitation. While the reported quality controls are reasonable, the observed LLM-human agreement suggests that some degree of annotation noise and model-induced bias may persist, and conclusions derived from the benchmark should be interpreted with this consideration in mind.
> >
> > I also appreciate the clarification that the scope of the paper is intentionally restricted to reading comprehension and question answering. I agree that extending the analysis to broader reasoning settings would be valuable future work, but I do not view this as necessary for acceptance given the stated scope of the paper.
> >
> > Overall, the rebuttal adequately addresses my primary concern regarding potential overinterpretation of the attention analysis. The remaining concerns mainly relate to dataset construction limitations and the scope of evaluation rather than the validity of the presented results.

---

### Review · Reviewer_6Ehw · 2026-05-18

**Summary Of Contributions:**

This paper studies the effect of emotional tone on the behavior of large language models in question answering. The authors first observe that model performance varies across contexts with different emotional valences. To analyze this phenomenon, the paper introduces a set of attention-geometry features, such as locality, center-of-mass distance, and related statistics, and uses these features to study the relationship between emotional tone, attention behavior, and QA accuracy. The paper further proposes a LoRA-based fine-tuning method that constructs an emotional latent subspace using SVD and applies a pairwise regularization loss to reduce emotion-conditioned representational drift. The intended contribution is to provide both a mechanistic analysis of emotion-induced performance differences and a training method for improving robustness under emotionally varied contexts.

**Audience:**

Yes

**Audience Explanation:**

The topic is relevant to the TMLR audience, since it is related to robustness, bias, interpretability, and representation analysis in large language models. The observation that emotional tone may affect QA performance is interesting, especially if the dataset is carefully controlled and emotionally balanced. The attention-geometry analysis may also be useful as a diagnostic tool for studying how input style changes model behavior.

However, the current paper does not yet present a fully convincing or well-integrated contribution. The attention-geometry findings, the emotional latent representation analysis, and the LoRA-based regularization method are each potentially interesting, but the paper does not organize them into a clear causal or methodological pipeline. As a result, the findings may interest some readers, but the current evidence and presentation are not strong enough to support the broader claims made by the paper.

**Broader Impact Concerns:**

There is no concerns.

**Claims And Evidence:**

No

**Claims Explanation:**

The paper provides evidence that emotional tone is associated with systematic changes in attention geometry, and that attention-derived features can predict QA correctness or emotion labels to some extent. These results suggest that emotional contexts may influence model behavior through internal processing patterns. However, the evidence does not fully support the stronger claims implied by the paper.

A main concern is that the paper relies on several implicit assumptions. The paper discusses multiple related but distinct phenomena, including emotion-induced performance variation, model-internal emotional bias, emotion-conditioned attention geometry, and emotional latent representations. These concepts are connected at a high level, but they are not equivalent. Showing that contexts with different emotional valences lead to different model performance does not directly establish that the model has intrinsic emotional bias. Similarly, showing that attention-derived features correlate with QA accuracy does not establish that attention geometry is the causal mechanism behind the performance gap.

Another concern is that the attention-geometry analysis and the proposed regularization method are not sufficiently integrated. The diagnostic part of the paper focuses on attention geometry, while the method operates on hidden-state representations through an SVD-derived emotional subspace. This creates a gap between the identified phenomenon and the proposed solution. The paper does not clearly show whether the LoRA regularizer reduces the attention-geometry disparities reported earlier, or whether changes in attention geometry mediate the improvement in QA performance.

The attention-geometry analysis is limited to correlation. Logistic regression and random forest classifiers can show that attention-derived features are predictive of accuracy or emotion labels, but this does not rule out alternative explanations. Factors such as passage length, answer position, topic distribution, lexical artifacts, rewriting style, or tokenization differences may also affect both attention statistics and model accuracy. Without stronger controls or intervention-based analysis, the mechanistic interpretation remains under-supported.

Finally, the proposed method appears to have limited methodological novelty. Constructing controlled emotional variants, extracting a latent concept subspace using SVD or PCA, and applying projection or consistency regularization is a common strategy in representation analysis, debiasing, and invariant learning. The paper applies this general framework to emotional QA, but it does not clearly distinguish the proposed method from existing subspace-based debiasing or representation-invariance approaches.

**Requested Changes:**

The authors should clarify the main target phenomenon. The paper discusses both context-induced emotional sensitivity and model-internal emotional bias, but these are related yet different claims. If the goal is to study model emotional bias, the authors should provide controlled counterfactual evidence; if the goal is to study emotional context effects, the claims should be framed more narrowly.

The authors should provide a clearer connection between the attention-geometry analysis and the representation-level regularization method. The proposed method uses an SVD-derived emotional latent subspace and a hidden-state consistency loss, while the diagnostic analysis focuses on attention-geometry features. The paper should show whether the proposed regularizer reduces the measured attention-geometry disparities and whether this reduction is associated with the reported performance improvement.

The authors should clarify the methodological novelty of the proposed approach. The current method is closely related to standard debiasing and invariant-representation techniques, where controlled variants are used to identify a bias-related subspace and then regularize or remove that subspace during training. The paper should explain what is technically new beyond applying this general framework to emotional variation.

The paper should improve its organization. The current presentation mixes emotional context effects, model emotional bias, attention-geometry analysis, emotional latent representations, and LoRA-based regularization. These components should be separated more clearly, and the paper should explain how each component supports the main claim.

---

> ### Author Response · Authors · 2026-06-07
>
> We thank the reviewer for the thoughtful and detailed feedback. We agree that the paper studies several related phenomena, including emotion-conditioned performance variation, attention geometry, emotional latent representations, and emotion-aware regularization. We also agree that these concepts are distinct and should not be treated as interchangeable.
>
> **Clarifying the target phenomenon.** Our primary objective is to study emotion as a latent contextual factor that influences question-answering performance, rather than to establish the existence of a specific form of model-internal emotional bias. The core empirical observation is that emotional tone is associated with systematic variation in QA performance, even when the questions themselves are non-emotional. The attention-geometry analysis, emotional latent-space analysis, and regularization framework are intended as complementary perspectives on this phenomenon rather than as a single causal chain. We revised the manuscript to make this framing more explicit and softened any language that could be interpreted as implying causality.
>
> **Correlation versus causation.** We agree that the analyses in Section 3 are correlational rather than causal. Our intention was not to claim that attention geometry is the causal mechanism underlying the observed performance differences. Throughout Section 3, we deliberately use correlational language such as "associated with," "correlated with," and "co-vary with." Rather than establishing causality, the attention analysis is intended as a diagnostic characterization of how emotional tone is reflected in the model's internal processing. We agree that intervention-based experiments would be required to establish causality and clarified this distinction in the revised manuscript. Our goal was to demonstrate that attention geometry contains substantial information about emotional tone and QA accuracy, not that attention is the unique explanatory factor.
>
> **Connection between attention geometry and the regularization framework.** We appreciate this observation. The attention analysis and the regularization framework operate at different levels of abstraction. The former characterizes emotion-conditioned processing differences through attention statistics, while the latter operates directly on hidden-state representations, where LoRA updates naturally act. Our intention was not to claim that the regularizer specifically corrects the attention-geometry effects identified in Section 3. Rather, the attention analysis motivates the broader observation that emotional expression is associated with systematic internal processing differences, while the regularizer attempts to constrain one representation-level manifestation of these differences. We agree that directly measuring whether emotional regularization reduces the reported attention disparities would be an interesting extension and clarified the scope of the current claims.
>
> **Methodological novelty.** We agree that the individual components of the regularization framework are related to broader ideas in representation learning, invariant learning, and debiasing. The intended contribution is not the introduction of a fundamentally new optimization primitive, but rather the application of emotional latent-space structure to the problem of emotion-conditioned reading comprehension. We view the novelty as the combination of emotion-conditioned reasoning analysis, the AURA-QA benchmark, attention-geometry characterization, and emotional latent-space regularization within a unified framework. We revised the manuscript to more clearly communicate this positioning.
>
> **Organization and presentation.** We appreciate the suggestion and agree that the relationships among the different components of the paper can be communicated more clearly. We revised the presentation to more clearly separate the empirical findings, diagnostic analyses, and regularization framework.

---

> > ### Comment · Reviewer_6Ehw · 2026-06-11
> > **comment on the rebuttal**
> >
> > The authors position the main contribution as a unified framework combining the AURA-QA benchmark, attention-geometry analysis, emotional latent representations, and representation-level regularization. However, the rebuttal clarifies that the attention analysis and latent-space regularization are intended as complementary perspectives rather than as a single causal or methodological pipeline. This clarification weakens the claim of a unified framework, since the paper does not establish a clear connection between the emotional representations and the reported attention-geometry metrics.
> >
> > The proposed method operates on hidden-state representations through an SVD-derived emotional subspace and a representation-level regularization objective. In contrast, the attention-geometry features do not guide the construction of the emotional subspace, define the regularization objective, or evaluate whether the intervention succeeds. The benchmark, attention analysis, and regularization method therefore remain largely separate components connected by the same research topic, which makes the paper appear as a collection of related findings rather than a complete framework.
> >
> > The authors should either provide additional experiments connecting representation-level changes to the reported attention effects or narrow the contribution to emotional representations and representation-level regularization. More detailed analyses of subspace quality, semantic-information preservation, layer-wise representation alignment, and the relationship between representation changes and QA performance would make the contribution more focused and convincing.

---

### Review · Reviewer_QWzt · 2026-05-27

**Summary Of Contributions:**

This paper presents the observation that varying emotional tones in data introduce bias into the analytical and reasoning capabilities of Large Language Models (LLMs). To investigate this mechanism, the authors detail three main components: First, they use attention geometry to quantitatively analyze how emotion shifts the model's internal attention allocation. Second, they introduce AURA-QA, a human-authored dataset balanced across nine emotions to mitigate the class imbalance present in standard benchmarks. Third, they propose an emotional regularization framework using a LoRA-based training objective. This method applies orthogonal projection via Singular Value Decomposition (SVD) to decouple emotional variables from semantic representations, aiming to stabilize factual reasoning across different affective contexts.
A primary strength of this work is its conceptual framework, treating emotion as a latent confounding variable that disrupts logical reasoning rather than focusing on emotional intelligence. The geometric metrics and the balanced AURA-QA dataset provide a controlled method for measuring affective interference. However, the proposed regularization method lacks generalizability. Empirical results show that the framework degrades in-domain performance on AURA-QA itself and shows little improvement on standard datasets like TweetQA. Thus the author's assumption, emotion and factual semantics exist in strictly orthogonal subspaces, is not universally correct. Removing emotional vectors can inadvertently strip away critical semantic information. Furthermore, while the dataset uses human text, relying on LLM consensus for labeling and QA generation risks embedding the synthetic artifacts the authors attempt to measure.

**Audience:**

Yes

**Audience Explanation:**

Its central contribution is the establishment of a standardized benchmark to evaluate the generalizability of LLMs across different emotional data types. This field is in need of further research.

**Broader Impact Concerns:**

Because AURA-QA is constructed from Project Gutenberg texts, the authors should acknowledge the risk of propagating historical social biases (such as outdated gender or racial stereotypes) inherently present in older public-domain literature, particularly when associating these texts with distinct emotional states.

**Claims And Evidence:**

Yes

**Claims Explanation:**

The claims are partially supported. To be exact, the analytical claims regarding attention geometry (Section 3) and the statistical rigor of the AURA-QA dataset (Section 4) are supported by clear and convincing empirical evidence. However, the central claim in the abstract and conclusion that the emotional regularization framework yields 'consistent gains' is strongly contradicted by the authors' own data in Table 6. The evidence shows that the method is brittle, failing to improve performance on TweetQA and significantly degrading in-domain performance on their newly proposed AURA-QA dataset. The authors must tone down their claims of generalization and explicitly address the negative results.

**Requested Changes:**

Critical changes:
1. Revise Claims of "Consistent Gains" and Generalization.
- The abstract and conclusion explicitly claim that the emotional regularization framework yields "consistent gains" across both emotionally varying and non-varying datasets. However, Table 6 shows negative or negligible results on Natural Questions, TweetQA, and significantly degrades in-domain performance on AURA-QA.
2. Clarify the LLM-Mediated Artifacts in AURA-QA
- The dataset relies on LLMs (LLaMA, Gemma, Qwen) for emotion verification and QA generation, testing LLMs on these LLM-generated distributions inevitably introduces synthetic artifacts. The authors must add a robust limitation statement discussing how evaluating models on data generated by their peers might skew the observed reasoning performance and mask true human-level comprehension gaps.

Strengthen suggests:
1. Expand on the TweetQA vs. FriendsQA Discrepancy.
- You can include a qualitative analysis explaining the performance discrepancy between FriendsQA and TweetQA under the regularization framework. Discussing whether emotions in short, dense texts (tweets) are too tightly coupled with semantics to be cleanly separated—compared to situational dialogue—would provide valuable insight into the functional boundaries of the proposed method.
2. Address the Performance Degradation on AURA-QA
- The proposed regularization method causes a substantial performance drop (4-8%) on the authors' own balanced dataset, AURA-QA, compared to the baseline LoRA. You can provide a dedicated discussion in Section 7 explaining why the method fails on a perfectly balanced dataset. Acknowledge the risk of "over-regularization" or the flawed assumption that emotional and semantic subspaces are strictly orthogonal in all contexts.

---

> ### Author Response · Authors · 2026-06-07
> **Rebuttal**
>
> We thank the reviewer for the thoughtful feedback and for recognizing the value of the attention-geometry analysis and the AURA-QA benchmark.
>
>
> **Claims of "consistent gains" and generalization.** We agree that the phrasing in the abstract and conclusion was imprecise and have revised both sections to more accurately characterize the results. Our intent was not to claim that emotional regularization improves every individual train/test configuration, but rather that it improves robustness on average across models and datasets. In Table 6, the average improvement relative to the baseline is positive for every training dataset and model, with an overall average gain of approximately 2.5%. While there are individual test datasets and train/test configurations where the framework does not yield improvements, the average effect across all (train, test, model) combinations remains positive. Moreover, many of the largest improvements occur in cross-dataset evaluation settings, suggesting that the regularizer can be particularly beneficial when models are evaluated under distribution shift. We agree, however, that the current wording may overstate the uniformity of the gains and have revised the abstract and conclusion to more clearly distinguish average robustness improvements from per-setting performance.
>
>
> **LLM-mediated artifacts in AURA-QA.** We agree that LLM involvement in emotion verification and QA generation introduces the possibility of synthetic artifacts. However, we note that the underlying passages themselves are human-authored and sourced from Project Gutenberg. In addition, we employed a multi-model consensus procedure (LLaMA, Gemma, and Qwen) together with human validation studies to reduce model-specific artifacts. The resulting dataset achieves 87.6% human validity for generated QA pairs, and passage-level LLM-human agreement approaches human-human agreement when restricted to unanimous LLM decisions. Our intention was therefore not to eliminate all possible generation artifacts, but rather to construct a scalable benchmark with stronger quality controls than single-model generation. We agree that this limitation merits careful consideration. These considerations are discussed in the limitations section, and we have added additional language there to more explicitly acknowledge the possibility of LLM-induced artifacts and the associated implications for interpreting benchmark results.
>
>
> **TweetQA versus FriendsQA.** We appreciate this suggestion. We note that Appendix F already contains an analysis of the relationship between latent-space alignment and regularization effectiveness across datasets. In particular, FriendsQA exhibits a roughly monotonic relationship between alignment and performance gains, whereas TweetQA displays a substantially more non-monotonic response. One possible explanation is that TweetQA consists of shorter, denser, and often irony-laden texts in which emotional expression is more tightly coupled to semantic content, making emotion-semantic separation more difficult. We emphasize that this analysis is already included in Appendix F and provides a possible explanation for the differing behavior of these datasets.
>
>
> **Performance degradation on AURA-QA.** We agree that the AURA-QA in-domain results warrant additional discussion. While emotional regularization generally improves out-of-domain performance when trained on AURA-QA, the same benefit does not consistently translate to the in-domain setting. We view this as evidence of a robustness-specialization tradeoff rather than a failure of the overall approach. Because AURA-QA is intentionally balanced across emotions, it reduces the emotional distribution shift that motivates the regularizer. Under these conditions, enforcing emotion-invariant representations may suppress information that remains useful for the source task, leading to a form of over-regularization. We expanded the discussion in Section 7 to better characterize this tradeoff and clarify that the method should not be interpreted as assuming perfect orthogonality between emotional and semantic information.